# Inferring latent temporal progression and regulatory networks from cross-sectional transcriptomic data of cancer samples

**Xiaoqiang Sun**[1,2]*, **Ji Zhang**[3], **Qing Nie**[4]*

**1** Key Laboratory of Tropical Disease Control, Chinese Ministry of Education; Zhongshan School of Medicine, Sun Yat-sen University, Guangzhou, China, **2** School of Mathematics, Sun Yat-sen University, Guangzhou, China, **3** State Key Laboratory of Oncology in South China, Collaborative Innovation Center for Cancer Medicine, Sun Yat-sen University Cancer Center, Guangzhou, Guangdong, China, **4** Department of Mathematics and Department of Developmental & Cell Biology, NSF-Simons Center for Multiscale Cell Fate Research, University of California Irvine, Irvine, California, United States of America

\* sunxq6@mail.sysu.edu.cn, xiaoqiangsun88@gmail.com (XS); qnie@uci.edu (QN)

## Abstract

Unraveling molecular regulatory networks underlying disease progression is critically important for understanding disease mechanisms and identifying drug targets. The existing methods for inferring gene regulatory networks (GRNs) rely mainly on time-course gene expression data. However, most available omics data from cross-sectional studies of cancer patients often lack sufficient temporal information, leading to a key challenge for GRN inference. Through quantifying the latent progression using random walks-based manifold distance, we propose a latent-temporal progression-based Bayesian method, PROB, for inferring GRNs from the cross-sectional transcriptomic data of tumor samples. The robustness of PROB to the measurement variabilities in the data is mathematically proved and numerically verified. Performance evaluation on real data indicates that PROB outperforms other methods in both pseudotime inference and GRN inference. Applications to bladder cancer and breast cancer demonstrate that our method is effective to identify key regulators of cancer progression or drug targets. The identified ACSS1 is experimentally validated to promote epithelial-to-mesenchymal transition of bladder cancer cells, and the predicted FOXM1-targets interactions are verified and are predictive of relapse in breast cancer. Our study suggests new effective ways to clinical transcriptomic data modeling for characterizing cancer progression and facilitates the translation of regulatory network-based approaches into precision medicine.

## Author summary

Reconstructing gene regulatory network (GRN) is an essential question in systems biology. The lack of temporal information in sample-based transcriptomic data leads to a major challenge for inferring GRN and its translation to precision medicine. To address the above challenge, we propose to decode the latent temporal information underlying cancer progression via ordering patient samples based on transcriptomic similarity, and

**Data Availability Statement:** The gene expression dataset of the bladder cancer was downloaded from the NCBI GEO database (GSE128192). The gene expression datasets as well as clinical

information of the breast cancer patients used for network prediction were downloaded from the NCBI GEO database (GSE7390). The microarray and ChIP-seq data used for network validation were downloaded from the NCBI GEO database (GSE40766, GSE40762, GSE62425, GSE2222, GSE58626 and GSE27830). The clinical gene expression data used for survival analysis were downloaded from the NCBI GEO database (GSE2990, GSE12093, GSE5327, GSE1456, GSE2034, GSE3494, GSE6532 and GSE9195). The gene expression RNAseq and phenotype information associated with the TCGA COAD dataset were downloaded from the UCSC Xena website (https://xena.ucsc.edu/) via the following links: https://tcga-xena-hub.s3.us-east-1.amazonaws.com/latest/TCGA.COAD.sampleMap/HiSeqV2.gz and https://tcga-xena-hub.s3.us-east-1.amazonaws.com/latest/TCGA.COAD.sampleMap/COAD_clinicalMatrix; The gene expression RNAseq and phenotype information associated with the TCGA SKCM dataset were also downloaded from the UCSC Xena website (https://xena.ucsc.edu/) via the following links: https://tcga-xena-hub.s3.us-east-1.amazonaws.com/latest/TCGA.SKCM.sampleMap/HiSeqV2.gz and https://tcga-xena-hub.s3.us-east-1.amazonaws.com/latest/TCGA.SKCM.sampleMap/SKCM_clinicalMatrix. A recent re-quantification of the LPS scRNA-seq dataset (GSE48968) was downloaded from the conquer database (http://imlspenticton.uzh.ch:3838/conquer/). The code for PROB is available at https://github.com/SunXQlab/PROB. The numerical data underlying graphs in the manuscript is available at S1_Data.xlsx in the Supporting Information.

**Funding:** XS was supported by grants from the National Natural Science Foundation of China (11871070, 11931019), the Guangdong Basic and Applied Basic Research Foundation (2020B151502120), the Fundamental Research Funds for the Central Universities (20ykzd20). QN was partially supported by a National Science Foundation grant DMS1736272, a Simons Foundation grant (594598), and a National Institute of Health grant U54CA217378. The funders had no role in study design, data collection and analysis, decision to publish, or preparation of the manuscript.

**Competing interests:** The authors have declared that no competing interests exist.

design a latent-temporal progression-based Bayesian method to infer GRNs from sample-based transcriptomic data of cancer patients. The advantages of our method include its capability to infer causal GRNs (with directed and signed edges) and its robustness to the measurement variability in the data. Performance evaluation using both simulated data and real data demonstrate that our method outperforms other existing methods in both pseudotime inference and GRN inference. Our method is then applied to reconstruct EMT regulatory networks in bladder cancer and to identify key regulators underlying progression of breast cancer. Importantly, the predicted key regulators/interactions are experimentally validated. Our study suggests that inferring dynamic progression trajectory from static expression data of tumor samples helps to uncover regulatory mechanisms underlying cancer progression and to discovery key regulators which may be used as candidate drug targets.

## Introduction

Inferring gene regulatory networks (GRNs) from molecular profiling of large-scale patient samples is of significance to identifying master regulators in disease at systems level [1]. Detecting the causal relationships between genes from biomedical big data, such as clinical omics data, has recently emerged as an appealing yet unresolved task, particularly for clinical purposes (e.g., diagnosis, prognosis and treatment) in the era of precision medicine [2].

Many methods have been developed for inferring GRNs from gene expression data [3]. The GRN inference methods can be grouped into at least four categories: Boolean network methods [4], ordinary differential equation (ODE) model-based methods [5], Bayesian network methods [6] and tree-based ensemble learning methods [7]. These methods mainly rely on two types of gene expression data, i.e., gene perturbation experiments [8,9] or time-course gene expression data [10]. Temporal changes in expressions, resulting from the interactions between genes, could potentially imply causal regulations. Meanwhile, a wealth of time-course transcriptomic data has been generated from the laboratory experiments. So temporal type of expression data is one of the most common assumptions based on which many GRN inference methods were designed [11].

However, the transcriptomic data of tumor samples often lack explicit temporal information [12]. In fact, large samples of time-course data are rarely available in clinical situations, at least for now, since longitudinal surveys are often challenging to conduct. In contrast, cross-sectional studies (i.e., a snapshot of a particular group of people at a given point in time) based on high-throughput molecular omics data are more prevalent due to their relative feasibility. As such, for cross-sectional transcriptomic data at population-scale, most of the current methods, such as Pearson correlation coefficient (PCC)-based methods [13], mutual information [14], regression methods [15] and machine learning methods [16], can only infer co-expressions or associations between genes. Moreover, although some correlation network-based methods have been used to identify disease-associated genes [17], it's hard to tell the causal drivers or regulatory roadmap underlying phenotypic abnormality in the absence of regulatory network information [18]. Therefore, the lack of temporal information in clinical transcriptomic data leads to a key challenge for inferring directed GRN and its translation to systems medicine.

Decoding temporal information that traces the underlying biological process from the cross-sectional data is intriguing and enlightening to address the above challenge. The sample similarity-based approach has shown great promise in recovering evolutionary dynamics in

evolution and genetics studies [19], for instance, phylogenetic trees based on microarray data [20] and genetic linkage maps based on genetic markers [21]. To this end, we propose that the latent temporal order of cancer progression status (i.e., latent-temporal progression) could be estimated from the cross-sectional data based on transcriptomic similarity between patient samples. Leveraging the latent-temporal ordering, we could represent the GRN as a nonlinear dynamical system. What's more, however, considering the technical variability or measurement error in the RNA-sequencing or microarray data (e.g., variations in sample preparation, sequencing depth and measurement noise and bias) [22,23], it's indispensably important to guarantee the robustness of the GRN inference.

In this study, we present PROB, a latent-temporal progression-based Bayesian method of GRN inference designed for population-scale transcriptomic data. To estimate the temporal order of cancer progression from the cross-sectional transcriptomic data, we develop a staging information-guided random walk approach to efficiently measure manifold distance between patients in a large cohort. In this way, the cross-sectional data could be reordered to be analogous to time-course data. This transformation enables us to formulate the GRN inference as an inverse problem of progression-dependent dynamic model of gene interactions, which is solved using a Bayesian method. The robustness of the estimates of regulatory coefficients is justified through mathematical analysis and simulations. Furthermore, applications to real data not only demonstrate better performance of PROB than other existing methods but also show good capacity of PROB in identifying key regulators of cancer progression or potential drug targets. The identified ACSS1 in bladder cancer and predicted FOXM1-targets interactions in breast cancer are both validated. In addition, we also discuss potential clinical applications of our method.

## Methods

### Ethics statement

The tumor tissues in this study were received from the operative resection of bladder cancer patients. The patients/participants provided their written informed consent to participate in this study. The studies involving human participants were reviewed and approved by the Ethics Committee of Sun Yat-sen University Cancer Center (approval no. GZR2018-131).

### Latent-temporal progression-based Bayesian (PROB) method to infer GRN

**Overview of PROB.**  PROB consists of two major components. First, to infer the latent temporal information of cancer progression from the cross-sectional data, a graph-based random walk approach was developed to quantitatively order patient samples (**Fig 1A and 1B**). More specifically, we defined a manifold distance between patients by analytically summing the transition probabilities over all random walk lengths to quantify temporal progression and the root of the progression trajectory was automatically identified with the aid of staging information. The quantitative reordering of the samples led to the recovery of the temporal dynamics of gene expression (**Fig 1C**). Second, a progression-dependent dynamic model was proposed to mechanistically describe the gene regulation dynamics during the above estimated temporal progression. To ultimately infer the GRN, the inverse problem in terms of parameter estimation of the dynamic model was transformed to a linear regression model which was solved using a Bayesian Lasso method (**Fig 1D**). Compared to the existing correlational network methods, PROB can infer causal GRNs with directed and signed edges from cross-sectional transcriptomic data.

**Temporal progression inference for cancer samples.**  We employ a similarity graph-based random walk approach to order patients along with the progression and to estimate the

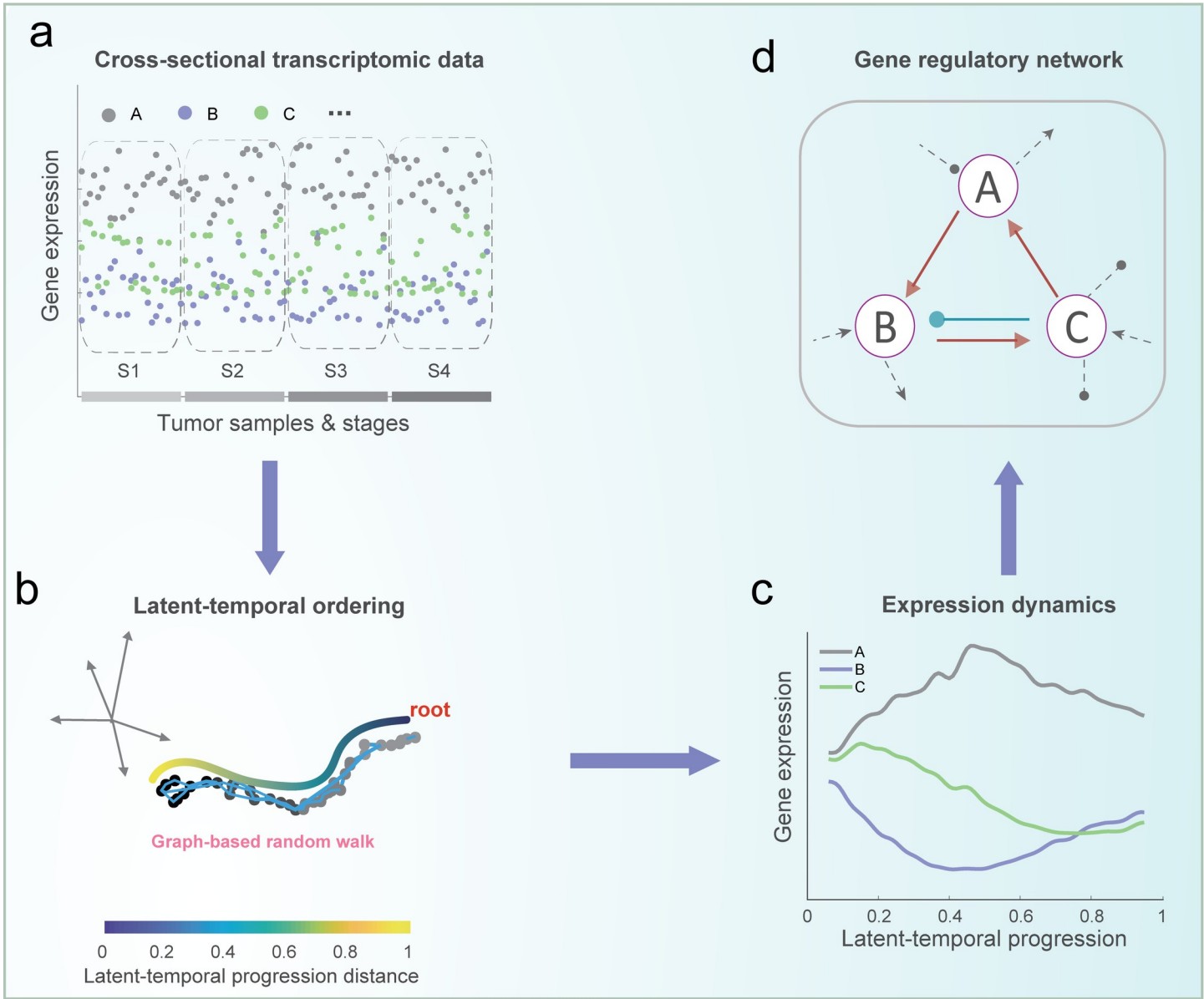

**Fig 1. Illustration of the PROB framework for inferring the causal gene regulatory network from cross-sectional transcriptomic data.** (**a**) Illustration of cross-sectional transcriptomic data, taking three genes (i.e., A, B, and C) as an example. Each sample was labeled with staging information (e.g., S1, S2, S3, and S4). (**b**) Similarity graph-based random walk approach for cancer progression inference. A scale-free temporal progression distance (TPD) is defined by analytically summing the transition probability between patients over all random walk lengths. Patients are thus ordered according to the TPD with respect to the root identified with the aid of staging information. (**c**) The expression dynamics of each gene according to the latent-temporal progression are then recovered. (**d**) A Bayesian Lasso method is developed to infer the causal GRN based on the temporal data of gene expression. Besides edge directions, PROB can also infer signs of the interactions (activation or inhibition), compared to the existing correlational network methods.

progression score for each patient, given the hypothesis that the similarity between patients can be measured by the patients' gene expression profiles and pathology information.

We first define a Gaussian similarity function for two patients, $x$ and $y$, as

$$S(x, y) = \exp(-\gamma \|T_x - T_y\|^2) \tag{1}$$

Where $T_x$ and $T_y$ are vectors used to represent the transcriptomic expression profiles of the respective patients and $\|T_x - T_y\|$ is the $L^2$ norm of $T_x - T_y$. The parameter $\gamma$ is determined as

$$\gamma = \frac{\omega_{xy}}{\varepsilon_x^2 + \varepsilon_y^2} \tag{2}$$

where $\omega_{xy}$ is a weight coefficient given by pathology information such as stage or grade, which is defined in this study as $\omega_{xy} = 1 + |G_x - G_y|$, with $G_x$ and $G_y$ representing grading or staging information (taking values of, for instance, 1, 2, 3, or 4) of the two patients $x$ and $y$, respectively. The parameter $\varepsilon_x$ is adaptive for each patient $x$ and is set as the patient's distance to the $\kappa$-th nearest neighbor. $S$ can be viewed as a stage-weighted and locally scaled Gaussian kernel.

To eliminate the effect of sampling density, we subsequently remold the above rotation-invariant kernel $S(x, y)$ into an anisotropic kernel $H(x, y)$,

$$H_{xy} = \frac{S(x, y)}{D(x)D(y)}, \tag{3}$$

by normalizing $S$ with a proxy for the sampling density of the data points,

$$D(x) = \sum_{y \in \Omega} S(x, y). \tag{4}$$

We next define a transition probability matrix $P$ whose elements are defined as

$$P_{xy} = E(x)^{-\frac{1}{2}} H_{xy} E(y)^{-\frac{1}{2}}, \tag{5}$$

where $E(x)$ is the row normalization of $H$, that is,

$$E(x) = \sum_{y \in \Omega} H_{xy}. \tag{6}$$

$P$ is a symmetric transition matrix [24,25]. $P_{xy}$ can be interpreted as the probability of transitioning from $x$ to $y$ (or from $y$ to $x$). The eigenvectors of $P$ can be referred to as diffusion components, and taken together, they constitute a modified version of the diffusion map [24,25], which extracts the topological structure of the high-dimensional data.

We then measure the transitions on all length scales between patients. The accumulated transition probability ($Q_{xy}$) of visiting $y$ from $x$ over random walk paths of all lengths is analytically calculated as

$$Q = \sum_{t=1}^{\infty} \tilde{P}^t = (I - \tilde{P})^\dagger - I \tag{7}$$

where $\tilde{P} = P - \psi_0 \psi_0^T$, and $\psi_0$ is the first eigenvector of $P$ (corresponding to eigenvalue 1). Since $\psi_0$ is associated with the steady state and contains no dynamic information [26], we subtract the stationary component $\psi_0 \psi_0^T$ from $P$, resulting in $\tilde{P}$. In this way, all the eigenvalues of $\tilde{P}$ are smaller than 1; hence, the above sum of infinite series is convergent. $(I - \tilde{P})^\dagger$ is the generalized inverse (or Moore-Penrose inverse) of $I - \tilde{P}$ [27].

We use $Q(x, \cdot)$ to represent the accumulated transition probability of visiting all points from $x$. Thus, $Q(x, \cdot)$ is a row in $Q$ and can be viewed as a feature representation for patient $x$. Therefore, we define a temporal progression distance (TPD) between two patients as

$$TPD(x, y) = \|Q(x, \cdot) - Q(y, \cdot)\|_{L^2}, \tag{8}$$

where $\|\cdot\|$ stands for the $L^2$ norm. We remark that TPD is a scale-free manifold distance and is computationally efficient due to the closed form expression of $Q$.

Given a patient $x$, the progression score with respect to the trajectory's root $x_0$ is $s = TPD$ $(x_0, x)$. Therefore, it is critical to determine the root sample in a large cohort for ordering the patients. We fulfill this task with the aid of the staging information of the patients: among all patients, the root of the trajectory should have the largest TPD to a patient with maximal stage (e.g., stage 4). That is, the root sample $x_0$ can be identified according to the following formula:

$$x_0 = \underset{x \in \{x_{\min}\}}{\operatorname{argmax}} TPD(x, x_{ref}), \tag{9}$$

where $x_{ref}$ is a randomly selected patient from the maximal grade subpopulation. The selection of $x_0$ was limited among patients with the smallest grade (i.e., $\{x_{\min}\}$) to eliminate potential influence of a few outliers in the data. The ordering of the progression scores quantifies the relative progression status and maps the patients into a smoothed temporal trajectory.

We remark that the incorporation of staging information into the Gaussian kernel and root identification could significantly improve the accuracy of temporal progression inference (see **S1 Fig** and Discussion section).

**Dynamical systems modeling.** Based on the mass action kinetics [28], the temporal regulation of gene expressions can be modeled using the following dynamical system,

$$\frac{dX_i(s)}{ds} = \sum_{j \neq i} a_{ij} X_i(s) \cdot X_j(s) - d_i X_i(s), (i = 1, \ldots, n) \tag{10}$$

where $X_i(s)$ represents the expression level of gene $i$ ($i = 1, \ldots, n$) in cancer with progression status $s$. $a_{ij}$ is the regulatory coefficient from gene $j$ to gene $i$ ($i = 1, \ldots, n; j \neq i$), and $d_i$ is the self-degradation rate of gene $i$. The details of model assumption and derivation are provided in **S1 Text**.

**Parameter estimation using Bayesian Lasso method.** Take $m+1$ points $S_i = s(r_i)$ from the smoothed progression trajectory $s(r)$, where $r_i = i/m$, $i = 0, 1, \cdots, m$. We approximate

$$\frac{dX_i}{ds}(s_k) \approx \frac{X_i(s_{k+1}) - X_i(s_k)}{s_{k+1} - s_k}$$

and denote

$$Y_{ik} = \frac{X_i(s_{k+1}) - X_i(s_k)}{s_{k+1} - s_k},$$

where $s_{k+1} - s_k$ is sufficiently small (since $m$ could be chosen large enough). Therefore, the above continuous model (i.e., Eq (10)) can be discretized and rewritten as

$$Y_{ik} \approx \sum_{j=1}^{n} a_{ij} X_i(s_k) \cdot X_j(s_k) - d_i X_i(s_k), (k = 0, 1, \cdots, m). \tag{11}$$

We then denote

$$Y_i = (Y_{i0}, \cdots, Y_{ik}, \cdots, Y_{im})_{1 \times (m+1)} \tag{12}$$

$$A_i = (a_{i1}, a_{i2}, \cdots, a_{in}, -d_i)_{1 \times (n+1)}, \tag{13}$$

and

$$X^{(i)} = \begin{bmatrix} X_1(s_0)X_i(s_0) & X_1(s_1)X_i(s_1) & \cdots & X_1(s_m)X_i(s_m) \\ X_2(s_0)X_i(s_0) & X_2(s_1)X_i(s_1) & \cdots & X_2(s_m)X_i(s_m) \\ \cdots & \cdots & \cdots & \cdots \\ X_n(s_0)X_i(s_0) & X_n(s_1)X_i(s_1) & \cdots & X_n(s_m)X_i(s_m) \\ X_i(s_0) & X_i(s_1) & \cdots & X_i(s_m) \end{bmatrix}_{(n+1) \times (m+1)} \quad (14)$$

Consequently, Eq (11) can be transformed into the following linear regression model:

$$Y_i = A_i X^{(i)} + \varepsilon_i, \ (i = 1, 2, \cdots, n), \quad (15)$$

where $\varepsilon_i = (\varepsilon_{i0}, \varepsilon_{i1}, \cdots, \varepsilon_{im})^T$ are the random effects with each $\varepsilon_{ik} \sim N(0, \sigma_i^2)$, $(k = 0, 1, \cdots, m)$.

We then use an adapted Bayesian Lasso method to infer the posterior distribution over the coefficients in each $A_i$. We assume that the conditional prior distribution of $A_i | \sigma_i^2, \lambda_i$ is the Laplace (double exponential) distribution with a mean of 0 and scale $\frac{\sigma_i}{\lambda_i}$, that is,

$$\pi(A_i | \sigma_i^2, \lambda_i) = Lap\left(0, \frac{\sigma_i}{\lambda_i}\right), \quad (16)$$

where $\lambda_i$ is the fixed lasso shrinkage parameter, which is set to 1. The prior distribution of $\sigma_i^2$, $\pi(\sigma_i^2)$, is usually assumed to be an inverse gamma, with the probability distribution function

$$f(x; A, B) = \frac{x^{-A-I \cdot e^{-1/xB}}}{(\Gamma(A)B^A)} \quad (17)$$

where A and B determine the shape and scale, respectively, of the inverse gamma distribution.

Using Bayes' rule, we formulate the joint posterior distribution of $A_i$ and $\sigma_i^2$ as follows:

$$\pi(A_i, \sigma_i^2 | Y_i, X^{(i)}) \propto \pi(A_i | \sigma_i^2, \lambda_i) \cdot \pi(\sigma_i^2) \cdot \ell(A_i, \sigma_i^2 | Y_i, X^{(i)}), \quad (18)$$

with $\ell(A_i, \sigma_i^2 | Y_i, X^{(i)})$, the data likelihood, given by

$$\ell(A_i, \sigma_i^2 | Y_i, X^{(i)}) = \prod_{k=0}^{m} \phi(Y_{ik}; A_i X_k^{(i)}, \sigma_i^2), \quad (19)$$

where $X_k^{(i)}$ is the (k+1)-th column of $X^{(i)}$, $(k = 0, 1, \cdots, m)$, and $\phi(Y_{ik}; A_i X_k^{(i)}, \sigma_i^2)$ is the Gaussian probability density with mean $A_i X_k^{(i)}$ and variance $\sigma_i^2$ evaluated at $Y_{ik}$.

The Markov chain Monte Carlo (MCMC) algorithm with Gibbs sampling updates is employed to estimate the marginal distribution of each parameter. A directed edge from gene $j$ to gene $i$ could be determined to be present if the 95% credible interval (CI) of the parameter estimates of $a_{ij}$ does not contain zero, otherwise absent.

**Mathematical analysis.** Considering the technical variability or measurement error in the transcriptomic data [22,23], it is important to examine the robustness of the method with respect to the perturbation in latent-temporal progression. To this end, we present the following theorem.

*Theorem 1.* Assume there are two trajectories of latent-temporal progression $s(r)$ and $\tilde{s}(r)$ with the same root, $r \in I = [0,1]$. Define $\|\tilde{s} - s\|_{L^2} = (\int_I |\tilde{s} - s|^2 dr)^{1/2}$. If $(X_i(s), a_{ij})$ and $(X_i(\tilde{s}), \tilde{a}_{ij})$ both satisfy the equations of progression-dependent dynamic model, i.e.,

$$\frac{dX_i(s)}{ds} = \sum_{j \neq i} a_{ij} X_i(s) \cdot X_j(s) - d_i X_i(s), \ i = 1, \cdots, n,$$

$$\frac{dX_i(\tilde{s})}{d\tilde{s}} = \sum_{j\neq i}\tilde{a}_{ij}X_i(\tilde{s})\cdot X_j(\tilde{s}) - \tilde{d}_iX_i(\tilde{s}), \; i = 1,\cdots,n,$$

then we have

$$\lim_{\|\tilde{s}-s\|\to 0}\sum_{i,j=1}^{n}|\tilde{a}_{ij} - a_{ij}|^2 = 0.$$

The proof of the above theorem is provided in **S2 Text**.

Based on the spectral graph theory [24,25], the above manifold distance (TPD) is noise-resistant, so the variation in the progression trajectory (i.e., $\tilde{s} - s$) should be small given moderate perturbations (as illustrated below). Consequently, **Theorem 1** then implies that the corresponding estimates of $[a_{ij}]_{n\times n}$ should vary minimally. Therefore, the above theorem theoretically guarantees the consistency and robustness of the estimates of the regulatory coefficients. In addition, the Bayesian Lasso method adopted by PROB further ensures a robust implementation of GRN inference.

A corollary of the above theorem is that the mapping $s\mapsto[a_{ij}(s)]_{n\times n}$ defined by Eq (10) is continuous under certain appropriate metric. More specifically, for two trajectories $s$ and $\tilde{s}$, if the difference between the two inferred regulatory coefficients $[a_{ij}(s)]_{n\times n}$ and $[a_{ij}(\tilde{s})]_{n\times n}$ is significantly larger than 0, then the difference between $s$ and $\tilde{s}$ should not be arbitrarily small. This implies that if the inferred regulatory networks for two progressions are largely different, then the two progressions should have different trajectories and thus distinct clinical outcomes. Hence, Theorem 1 also suggests that GRN-based signatures may be used for predicting or controlling cancer progression.

**Computational algorithm.** The algorithm to infer progression trajectory and GRN is presented below. The implementation of PROB is described in **S3 Text**.

*Algorithm 1*. pseudo-code of PROB

| | |
|---|---|
| 1: | **Input**: data = $[T, G]$. $T$, transcriptomic expression matrix; $G$, stage vector. |
| 2: | Stage-weighted Gaussian kernel: $S(x,y) = exp(-\gamma\|T_x - T_y\|^2)$ and $\gamma = \frac{1+|G_x-G_y|}{\varepsilon_x^2+\varepsilon_y^2}$. |
| 3: | Normalization of $S$ : $H_{xy} = \frac{S(x,y)}{D(x)D(y)}$. |
| 4: | Transition probability: $P_{xy} = E(x)^{-\frac{1}{2}}H_{xy}E(y)^{-\frac{1}{2}}$. |
| 5: | Accumulated transition probability: $Q = (I - (P - \psi_0\psi_0^T))^\dagger - I$; |
| | $\psi_0$ is the first eigenvector of $P$. |
| 6: | TPD function: $TPD(x,y) = \|Q(x,\cdot) - Q(y,\cdot)\|_{L^2}$. |
| 7: | Trajectory root: $x_0 = \underset{x\in\{x_{\min}\}}{\arg\max} TPD(x, x_{ref})$; $x_{ref}\in\{x: G_x = max(G)\}$. |
| 8: | Progression score: $s = TPD(x, x_0)$. |
| 9: | **For** $i = 1$ **to** $n$ **do** |
| | $Y_{ik} \triangleq \frac{X_i(s_{k+1})-X_i(s_k)}{s_{k+1}-s_k}$ |
| | $A_iX^{(i)} \triangleq \sum_{j=1}^{n}a_{ij}X_i\cdot X_j - d_iX_i$ |
| | $A_i = \text{BayesianLasso}(X^{(i)}, Y_i)$ |
| | **End** |
| 10: | **Output**: posterior distributions of $a_{ij}$, confidence matrix $CM$. |

## Benchmarking PROB with alternative methods of GRN inference

For tumor sample-based gene expression data, several methods have been developed to infer gene networks. Pearson correlation (PCOR) is often used to quantify gene coexpression. Mutual information (MI) measures non-linear dependency between genes and thus provides a natural generalization of the correlation. MI-based methods for GRN inference include ARACNe [29], CLR [30], and MRNET [31]. Another commonly used method for GRN inference based on gene expression data is multiple linear regression LASSO method [32], which assumes sparse network structure and is feasible for high-dimensional data. Ensemble learning methods, such as GENIE3 (a tree-based ensemble learning method [16]), have been developed to infer gene regulatory relationships by viewing GRN reconstruction as a classification problem. In addition, we also included some GRN inference methods recently developed for scRNA-seq data into benchmarking analysis, since scRNA-seq data is also cross-sectional type. Such methods include SCODE [33] that uses ordinary differential equations model and LEAP [34] that constructs gene co-expression networks by using the time delay involved in the estimated pseudotime of the cells. SINCERITIES [35] is designed for time-stamped scRNA-seq data but requires at least 5 time points, so it is not applicable for the following benchmarking dataset as well as the tumor sample-based transcriptomic data.

In this study, we compared the accuracy of PROB with that of PCOR, ARACNe, CLR, MRNET, Lasso, GENIE3, SCODE and LEAP based on a real scRNA-seq data of dendritic cells (DCs) (GSE41265 [36]). The cells were stimulated with LPS and sequenced at 1, 2, 4, and 6h after stimulation. Only wild type cells ($n = 479$) without Stat1 and Ifnar1 knockout were chosen for analysis. We choose this DC dataset for benchmarking because regulatory potential between 23 TFs in the DCs has been determined via a high-throughput Chromatin Immuno-Precipitation (HT-ChIP) method [37]. The AUC of ROC was used to assess and compare the prediction accuracies of the above methods.

In addition, we collected a set of known regulators and targets [38] to test whether PROB could correctly distinguish outgoing regulations of different genes. To this end, we defined an outgoing causality score (OCS) for gene $i$ in cell $k$ as follows: $OCS_i^k = \sum_{j=1}^{n} m_{ji} X_j^k X_i^k$, where $m_{ji}$ is the absolute value of mean of the posterior distributions of $a_{ji}$, $X_i^k$ is the expression level of gene $i$ in cell $k$. The OCS is defined in accordance of the Eq (10) based on the mass action kinetics and quantifies the outgoing regulatory potential of a give gene. We then compared the distributions of OCS values of 6 regulators and that of 28 targets using the above DC dataset. The Wilcoxon rank-sum test (one-tailed) $p$ value was calculated to assess statistical significance.

## Application to a dataset of bladder cancer

We applied PROB to a dataset of bladder cancer patients that includes 84 cases of conventional UCs and 28 cases of SARCs which were profiled by Illumina HumanHT-12 DASL Expression BeadChips (GSE128192 [39]). The temporal progression inference was performed to quantitatively order samples based on the whole gene expression profile with UC samples and SARC samples labeled by 1 and 2 respectively. To reconstruct epithelial-to-mesenchymal transition (EMT) regulatory networks, we collected 44 representative genes of TGFB1 pathway, RhoA pathway, p53 pathway, p63 pathway and EMT transcriptional regulators (**S1 Table**) [39]. The UC network and SARC network were reconstructed based on the ordered expression data of the above 44 genes in UC samples and SARC samples respectively. The UC-specific network and SARC-specific network were then constructed by extracting edges that were unique to UC network and SARC network respectively. The out-degree values for each node in the two networks were calculated to prioritize key regulator genes.

## Application to a dataset of breast cancer

We applied PROB to a microarray dataset of breast cancer (GSE7390 [40]) to identify key regulator genes with prognostic role in cancer progression. We identified the hub gene in the GRN based on an eigenvector centrality measure according to singular value decomposition method [41]. Denote the mean of the posterior distributions of $a_{ij}$ as $m_{ij}$, and $M = (m_{ij})_{n \times n}$. We subject $M$ to singular value decomposition. We calculated the principal eigenvector of $M \cdot M^T$ and denoted it $H = (h_1, h_2, \ldots, h_n)$. The hub score of node $i$ was defined as $h_i$. The gene with greatest hub score was identified as a hub gene for further analysis and validation.

## Validation of the role of ACSS1 in bladder cancer

*Antibodies and reagents*. Anti-β-actin Mouse mAb (1:1000, 0101ES10, Yeasen), anti-E-Cadherin Mouse mAb (1:1000, #14472, CST), anti-ACSS1 Rabbit mAb (1:1000, 17138-1-AP, Proteintech), Goat Anti-Rabbit IgG (H+L) (1:10000, 33101ES60, Yeasen), Goat Anti-Mouse IgG (H+L) (1:10000, 33201ES60, Yeasen), Anti-Rabbit IgG-HRP kit (SV0002, Boster).

*Over-expression plasmids and siRNA transfection*. 5637 cells were placed in 24 wells plate and transfected with the lentiviral vectors pTSB-CMV-puro and SiRNA against ACSS1 reaching 70%-80% confluence using Lipofectamine 2000 (Thermo Scientific) according to the manufacturer instructions. The SiRNA sequence used in this study are listed in **S2 Table**.

*RNA extraction and qPCR*. Total RNA was extracted by HiPure Total RNA Mini Kit (R4111-03, Magen) and the concentration was detected by ultramicrospectrophotometer (NanoDrop 2000, Thermo Fisher Scientific). RT-PCR was performed using PrimeScript RT Master Mix (DRR036A, TakaRa) and qPCR was performed by qPCR SYBR Green Master Mix (11198ES03, Yeasen) in Real-time quantitative PCR instrument (Q1000+, Long Gene). All the relative mRNA expression was normalized to GAPDH. The qRT-PCR primer sequence used in this study are listed in **S3 Table**.

*Western blotting*. Total protein was extracted by RIPA lysis buffer (JC-PL001, Genshare) with PMSF (1:100, 20104ES03, Yeasen). Standard western blot protocols were adopted. The band intensity of western blots was detected by BLT GelView 6000M. All the relative protein expression was normalized to β-actin.

*Immunohistochemistry*: All the tumor tissues were received from the operative resection of patients. The patients/participants provided their written informed consent to participate in this study. The studies involving human participants were reviewed and approved by the Ethics Committee of Sun Yat-sen University Cancer Center (approval no. GZR2018-131). The immunohistochemical analysis of the two markers including ACSS1 and E-Cadherin was performed. All the pathological sections were produced, scanned and analyzed by Leica Biosystems.

## Validation of the FOXM1 sub-network predictions

We validated the regulation of FOXM1 (a hub gene, see Results section) on the predicted targeted genes using multiple sets of gene expression data and ChIP-seq data that are publicly available.

To validate the expression changes of the predicted targeted genes following FOXM1 perturbation, we analyzed microarray gene expression data in MCF-7 cells that were treated with DMSO (control) or Thiostrepton (FOXM1 inhibitor) for 6 hours (GSE40762 [42]). The differential expression of the above 8 genes between control condition and FOXM1 inhibition condition was examined to test whether they were down-regulated after FOXM1 inhibition. The statistical significance was assessed using Wilcoxon rank sum test (one-tailed) p values.

To test whether FOXM1 binds to some of the predicted targeted genes, we used ChIP-seq data in both MCF-7 cell line (ER+) and MDA-MB-231 cell line (ER-) (GSE40762 [42]) to analyze binding of FOXM1. A standard procedure of the ChIP-seq analysis was performed for peak calling (S7 Text).

## Results

### Testing PROB with a synthetic dataset

To illustrate the function of PROB, we generated a set of synthetic cross-sectional expression data (S4 Text). For visualization purpose, we considered 6 genes in 100 cancer patients (S2A and S2B Fig). We first used PROB to infer temporal progression from the randomized sample-based data. The inferred latent-temporal progression was compared against the true progression (S2C Fig), showing that PROB faithfully recovered the true ordering of the samples (Spearman's rho = 0.9991). The gene expression dynamics along with latent-temporal progression (S2D Fig) exhibited a very similar profile to the original data (S2A Fig). Based on the inferred temporal data, PROB inferred a GRN using the Bayesian Lasso method (S2E Fig). The posterior distributions of the regulatory parameters against their true values show that the estimation was rather reliable (S3 Fig). An edge was determined by examining whether the 95% credible interval (CI) of the parameter estimates did not contain zero (S4 Text). S2F Fig further demonstrates the accuracy of PROB in terms of GRN inference. The area under curve (AUC) of receiver operating characteristic (ROC) could be calculated for the inferred network compared with the ground-truth network based on the $k$% CI that contained zero or not.

To verify the robustness of PROB to the measurement variability, we further tested PROB for datasets at different levels of variabilities (Fig 2). The gene expressions were randomly perturbed by using multiplicative Gaussian noises to simulate different levels of measurement variabilities in the data, resulting in a series of coefficient of variations (CVs) (i.e., 0%, 5%, 10% and 15% respectively) (Fig 2A). The IDs of the samples were randomized to mimic sample-based snapshots of gene expression data, but the staging information was retained for each patient. PROB was applied to infer the GRN for each dataset. The accuracy of GRN inference was evaluated using the AUC of the ROC, showing that PROB could strongly reduce bias in gene expression measurements (Fig 2B and 2C) and robustly reconstructed the GRNs (Fig 2D). Additional evaluation metrics were employed to verify the robustness of PROB against a series of variations in the data (with CVs ranging from 0% to 30%). The root mean square error (RMSE) and Spearman correlation coefficients were used to evaluate the accuracy of the temporal progression inference (S4A and S4B Fig). The accuracy, positive predictive value (PPV) and Matthews correlation coefficient (MCC) were used to evaluate the robustness of the GRN reconstruction (S4C–S4F Fig). In addition to the above Gaussian noises, we also tested the robustness of PROB against perturbations of multiplicative exponential noises generated from the exponential distribution with mean ranging from 0 to 0.3 (S5 Fig). The findings are consistent with the above results (Fig 2).

### Benchmarking PROB with other existing methods

We used a set of single cell RNA-seq (scRNA-seq) data (GSE48968 [36]) for benchmarking of GRN inference methods since our method can be naturally applied to stage-stamped or time-course scRNA-seq data and the ground-truth of the GRN is available in this case as described in the Methods section. The LPS-stimulated dendritic cells (DCs) were sequenced at 1, 2, 4, and 6h after stimulation. The capture time in the data was treated as an analogy to 'staging' information when using PROB. The estimated latent-temporal progression recapitulated the physical progression of cells with a high correlation to the capture times ($R^2$ = 0.851) (Fig 3A).

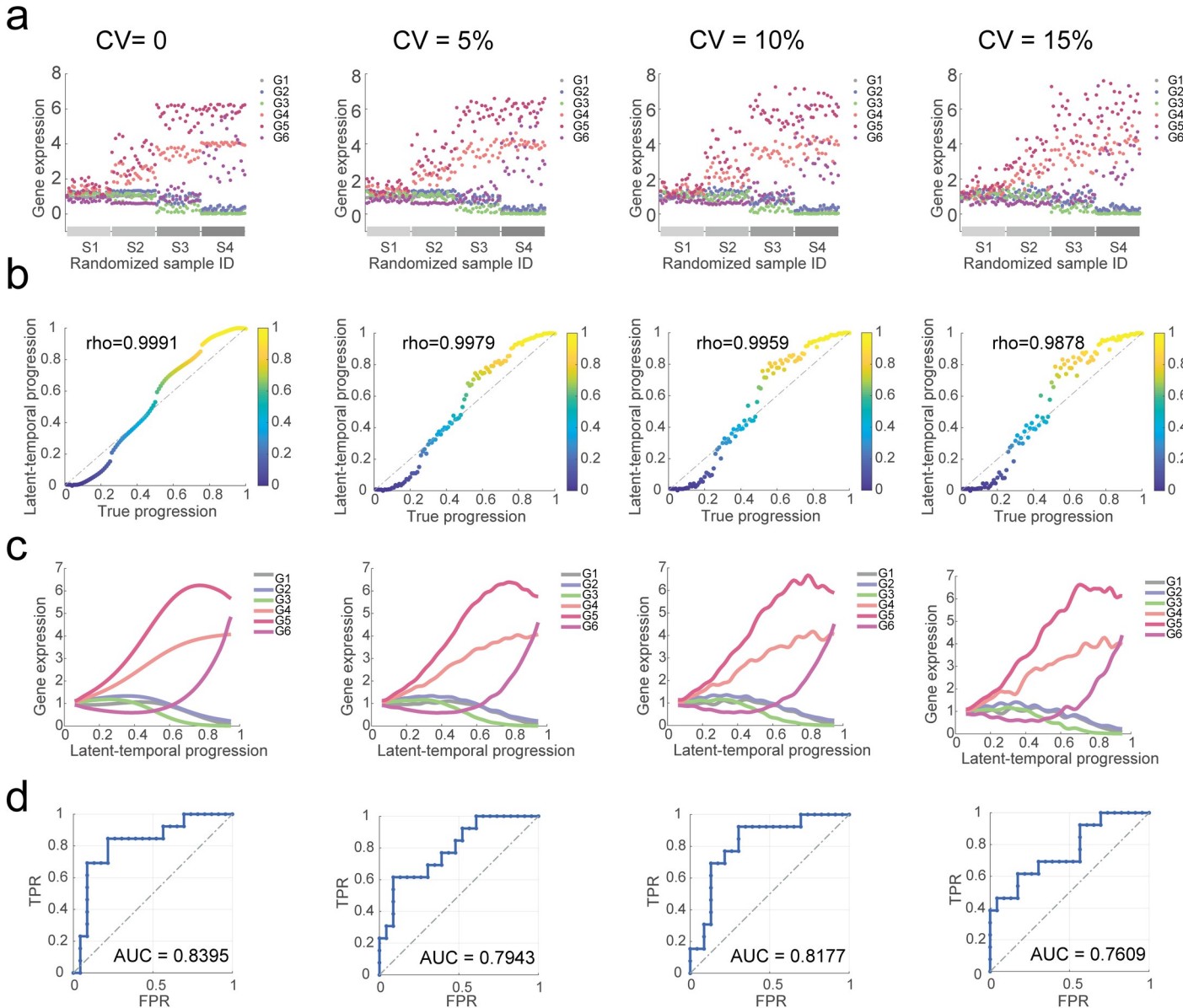

**Fig 2. Demonstrating robustness of PROB using synthetic datasets at different levels of variabilities.** A set of expression data for 6 genes in 100 cancer patients was simulated. Different levels of technical variabilities (with coefficient of variations (CVs) = 0%, 5%, 10% and 15% respectively) were introduced into the progression-dependent gene expression dynamics. (**a**) Simulated cross-sectional gene expression data. The sample IDs of the synthetic data were randomized and the staging information was retained. (**b**) Comparison of the inferred latent-temporal progression with the true progression in the synthetic dataset, evaluated using Spearman's rank correlation coefficient (rho). (**c**) Recovered gene expression dynamics according to inferred progression trajectory. (**d**) Accuracy of the GRN inference evaluated using the areas under curve (AUCs) of the ROCs.

We compared PROB with other pseudotime inference methods (Slice, Slicer, PhenoPath, Wishbone, PAGA, Monocole2, DPT, Tscan). PROB estimation achieved a highest correlation with the original physical capture times among all methods tested, evaluated using both Kendall Tau rank correlation coefficient (**Fig 3B**) and coefficient of determination $R^2$ (**S6 Fig**).

We next compared the accuracy of PROB with other existing GRN inference methods (e.g., PCOR, ARACNe, CLR, MRNET, Lasso, GENIE3, SCODE and LEAP) for cross-sectional data. A previous study measured binding region coverage scores for 23 TFs and thus quantified

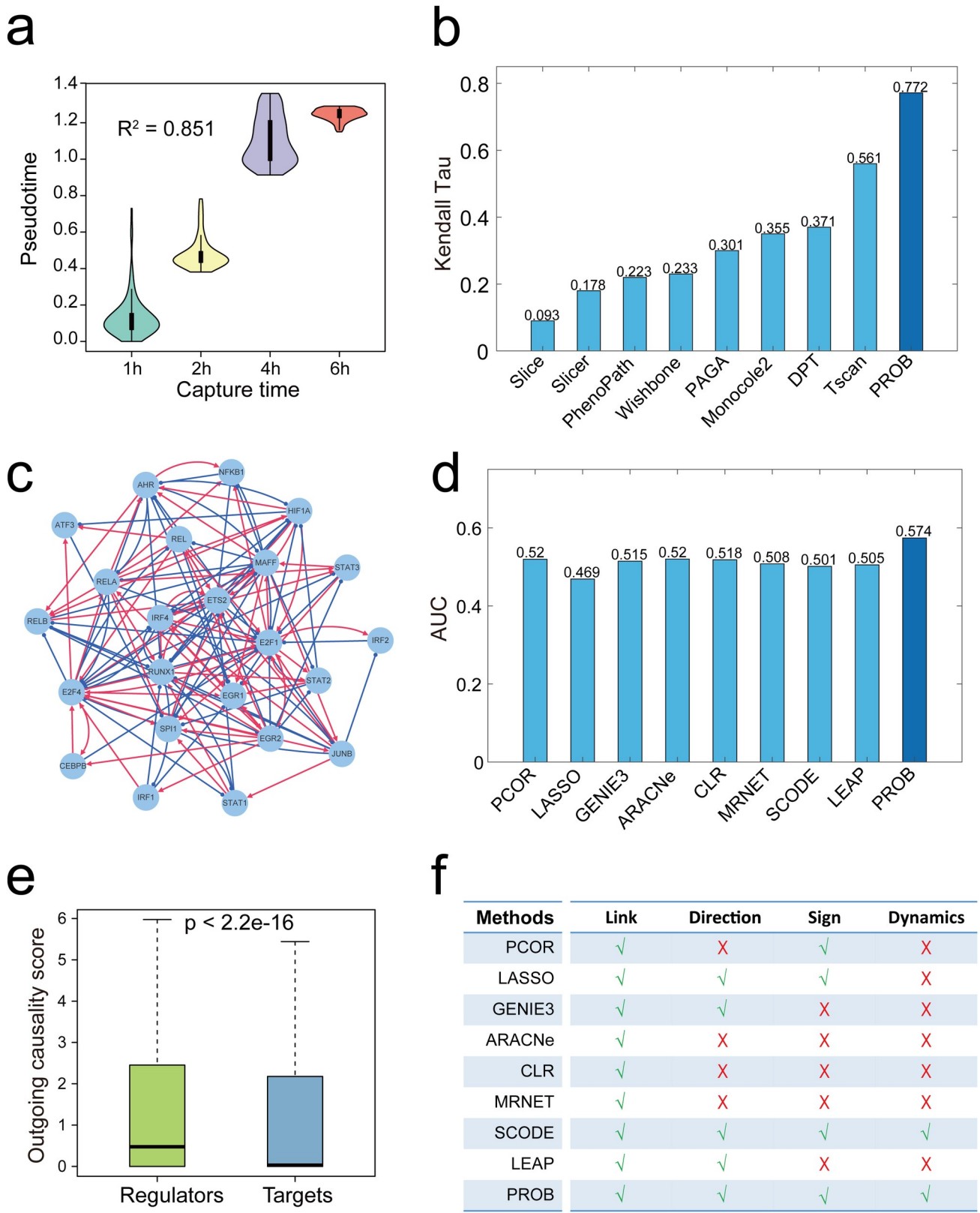

**Fig 3. Comparison of PROB with other existing pseudotime inference methods and GRN inference methods using a real dataset.** We employed a set of scRNA-seq data of dendritic cells (DCs) for benchmarking since the gold standard in this situation is available. The cells were sequenced at 1, 2, 4 and

6h after stimulation of LPS. (**a**) The estimated latent-temporal progression of cells recapitulated the real progression with $R^2 = 0.851$ to the capture times. (**b**) Benchmarking PROB with other pseudotime inference methods (Slice, Slicer, PhenoPath, Wishbone, PAGA, Monocole2, DPT, Tscan) evaluated by Kendall Tau and $R^2$ (S4 Fig). (**c**) a TF network inferred by PROB. (**d**) Benchmarking PROB with eight existing GRN inference methods (PCOR, LASSO, GENIE3, ARACNe, CLR, MRNET, SCODE and LEAP) based on an experimentally-defined TF network [37] evaluated by AUC of ROC. (**e**) PROB correctly revealed the ordering of the outgoing causality scores (on a log10 scale) for the known regulators and targets [38] on the DC scRNA-seq dataset. (**f**) Comparing properties of different methods in their capabilities of predicting network links, regulatory directions and signs as well as gene expression dynamics.

their regulatory potential in the DCs using a high-throughput Chromatin ImmunoPrecipitation (HT-ChIP) method [37]. A TF network was defined where an edge was viewed to be present if the coverage score between two TFs was greater than 0.3. We employed this network as a benchmark to compare the prediction accuracy of the network topologies inferred by PROB (**Fig 3C**) and other methods based on the above scRNA-seq data of DCs. The AUC values (**Fig 3D**) indicated that PROB outperformed the other existing methods.

Furthermore, we collected a set of known regulators and targets [38] to test whether PROB could correctly reveal the regulatory causality. To this end, we applied PROB to infer a GRN for 6 regulators and 28 targets based on the above DC scRNA-seq data and defined outgoing causality score (OCS) for each gene in the inferred network (see definition of OCS in the Methods section). The OCS values of regulators were much higher than that of targets (**Fig 3E**), suggesting that PROB faithfully revealed the ordering of the OCS values for the known regulators and targets on the analyzed dataset.

In addition, we summarized and compared the capabilities of the above methods in predicting gene regulatory links, directions, signs and expression dynamics (**Fig 3F**). Only PROB can simultaneously fulfill those four tasks in GRN inference.

## Reconstructing EMT regulatory networks during bladder cancer progression

Sarcomatoid urothelial bladder cancer (SARC) is a highly lethal variant of bladder cancer and has been reported to be evolved by the progression of the conventional urothelial carcinoma (UC) [39]. It has been demonstrated that the dysregulation of genes involved in the epithelial-to-mesenchymal transition (EMT) drives the progression of UC to SARC. To elucidate the dynamic change of the EMT regulatory network during the progression, here, we applied PROB to an expression dataset of bladder cancer containing 84 UC samples and 28 SARC samples (GSE128192). We collected 44 representative genes involved in several typical EMT-regulating pathways (**S1 Table**). The expression patterns of these genes were recovered along with the inferred temporal progression (**Fig 4A**).

We then applied PROB to reconstruct GRNs for UCs and SARCs, respectively, based on the ordered expression data of the above 44 genes. **Fig 4B** and **Fig 4C** show the UC-specific network and the SARC-specific network, respectively, suggesting rewiring of the EMT regulatory network during the progression of UC to SARC. The two networks were enriched with crosstalks between different pathways, indicating cooperative regulation of EMT by those pathways. PTPN12 and ACSS1 were found to have largest out-degree values in UC-specific network and SARC-specific network, respectively (**S1 Table**). Temporal dynamics of gene expression (**Fig 4D**) showed that ACSS1 and PTPN12 oscillated synchronously with CDH1 (coding gene of epithelial marker protein E-cadherin) at the early stage of UC development. However, at a later stage before transition to SARC, ACSS1 dramatically increased and PTPN12 decreased. Meanwhile, the decrease of CDH1 later on indicated a transition from epithelial to mesenchymal phenotype in SRACs, in consistent with changes in EMT score values (**Fig 4E**).

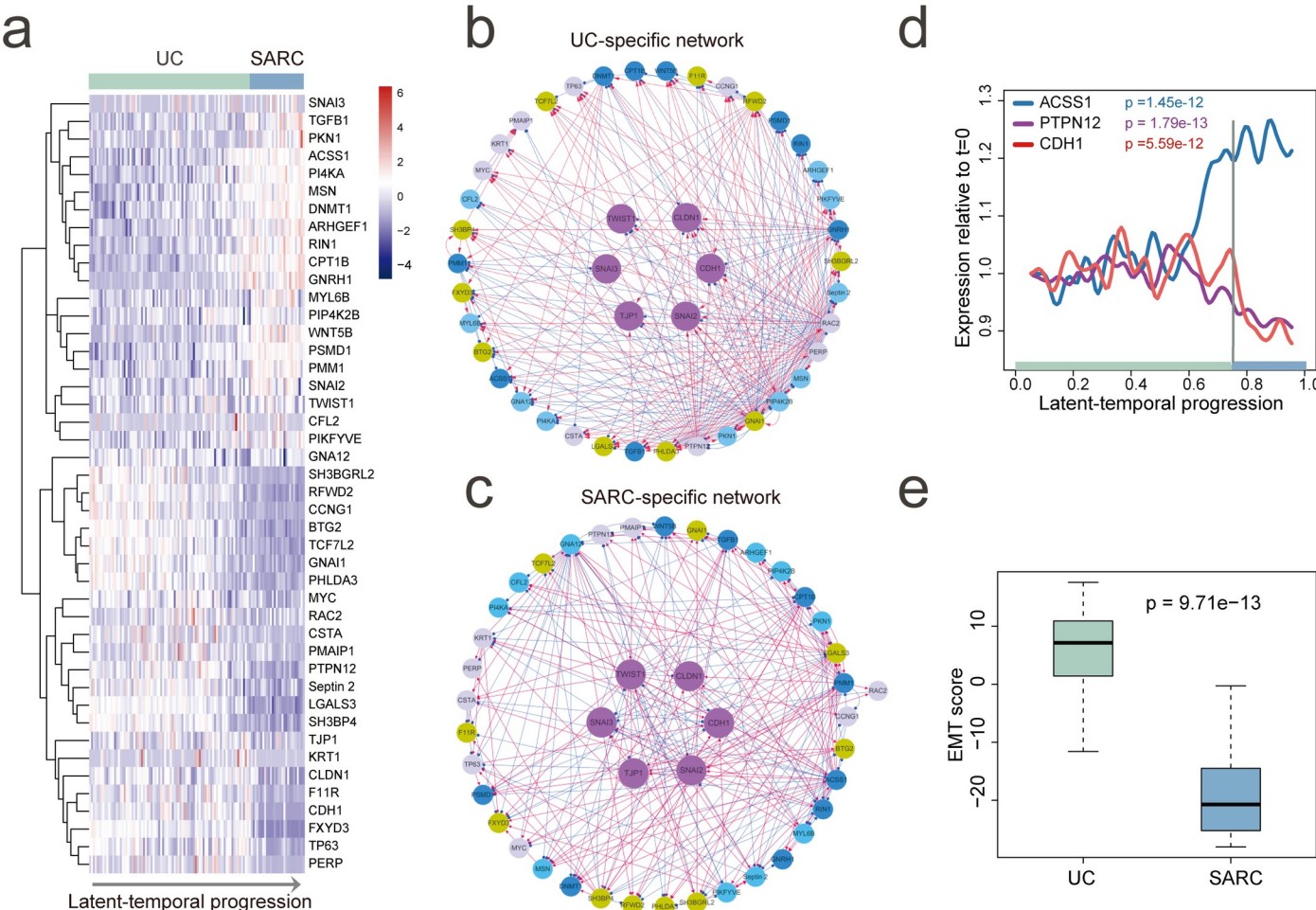

**Fig 4. Reconstructing EMT regulatory networks during bladder cancer progression.** (**a**) Expression patterns of the EMT regulatory genes along with the inferred latent-temporal progression of conventional urothelial carcinoma (UC) to aggressive sarcomatoid urothelial bladder cancer (SARC). (**b**) UC-specific network with edges unique to the UC network. (**c**) SARC-specific network with edges unique to the SARC network. Different colors of nodes in the network denote genes in different pathways (**S1 Table**). (**d**) Reconstructed expression dynamics of ACSS1, PTPN12 and CDH1. ACSS1 and PTPN12 have largest out-degree values in the UC-specific network and SARC-specific network, respectively. CDH1 is a marker gene of epithelial state during EMT. (**e**) A decrease in EMT score indicated a transition from epithelial to mesenchymal state during the progression of UC to SARC. The EMT score for each tumor sample was calculated as weighted sum of expression levels of 73 EMT-signature genes as introduced in [39]. Positive EMT score corresponds to the epithelial phenotype while negative score to mesenchymal phenotype. Wilcoxon rank sum test (one-tailed) p value was calculated to assess the statistical significance.

## Validation of the role of ACSS1 in EMT

The decrease in PTPN12 expression during the progression is consistent with the previous finding that the loss of PTPN12 promotes EMT process and cell migration [43]. Furthermore, our result suggests that the up-regulation of ACSS1 might play a crucial role in the bladder cancer progression by promoting EMT program. We managed to validate the role of ACSS1 in EMT during bladder cancer progression, which has not been reported previously. The overexpression of ACSS1 in the 5637 cell line resulted in a significant decrease in CDH1 expression level (**Fig 5A**), and ACSS1 knockdown by small interfering RNA led to significant increase in CDH1 expression level (**Fig 5B**). The consistent changes in CDH1 protein levels following ACSS1 overexpression and knockdown were also observed (**Fig 5C and 5D**). The numerical values of qPCR data and quantification of western blots were provided in **S1 Data**. These results confirmed that ACSS1 promoted EMT in bladder cancer cells. Furthermore, the

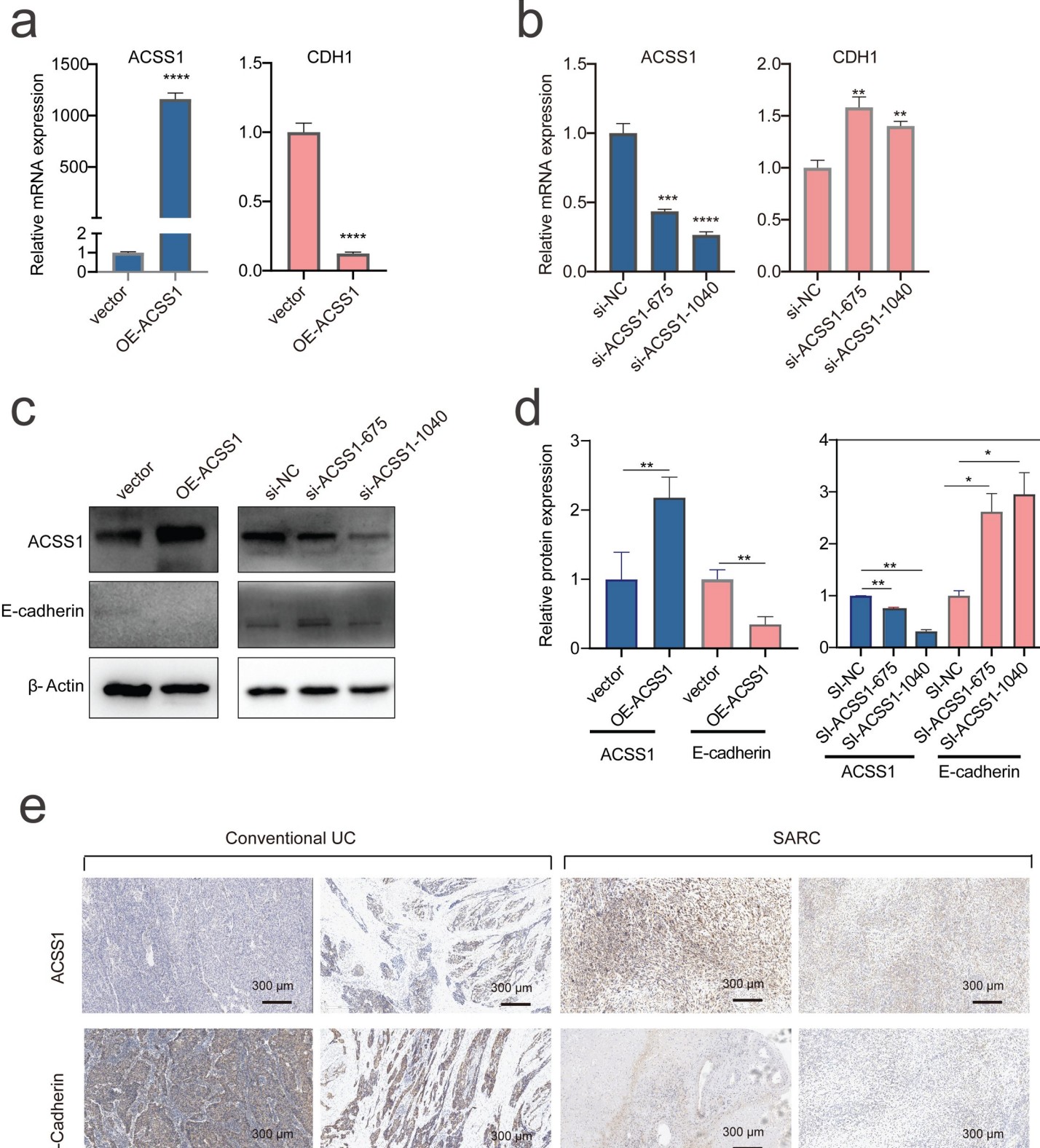

**Fig 5. Experimental validation of the predicted role of ACSS1 in EMT of bladder cancer.** (**a-b**) Expression levels of ACSS1 and CDH1 in 5637 cells when ACSS1 was overexpressed (a) and inhibited (b), measured by q-PCR. (**c**) Protein expression levels of ACSS1 and CDH1 in 5637 cells when ACSS1 was overexpressed or inhibited, measured by Western-blotting. (**d**) Quantification of the relative protein expressions. (**e**) Examples of immunohistochemical expression of ACSS1 and E-cadherin in

conventional UC and SARC. Statistical significance was assessed by student's t test. **P<0.01; ***P<0.001; ****P<0.0001. OE-ACSS1: overexpression of ACSS1; si-NC: small interfering RNA negative control; si-ACSS1: small interfering RNA targeting ACSS1.

immunohistochemical staining of patient samples (**Fig 5E**) revealed that conventional UC tumors showed focal retention of epithelial marker protein E-cadherin while SARC tumors showed focal retention of ACSS1, supporting the above estimated dynamics of ACSS1 and CDH1 during bladder cancer progression.

## Identifying key gene regulators underlying breast cancer progression

To test whether our approach could be used to identify key genes underlying cancer progression, we applied PROB to a set of microarray data of breast cancer patients (n = 196) with clinical information (GSE7390) (see details in **S5 Text**) [40]. Based on the expression data reordered by PROB, we investigated which genes were upregulated or downregulated over progression by using a trend analysis technique. Such genes are referred to as temporally changing genes (TCGs) in this study. The one hundred top TCGs were selected. A heatmap with hierarchical clustering (**Fig 6A**) showed that these 100 genes were clearly clustered into two groups: a descending group (purple) and an ascending group (blue). We investigated the enriched gene sets for the two groups of genes using GSEA software [44,45]. The descending genes were enriched in locomotion and movement of cell or subcellular component (**Fig 6B**, upper panel), and the ascending genes were mainly enriched in cell cycle and cell division processes (**Fig 6B**, lower panel).

We then inferred the regulatory network of the above 100 top genes (**Fig 6C**). Based on an eigenvector centrality measure (**S5 Text**), FOXM1 was identified as a most influential gene in the network. We verified significant associations between FOXM1 and the distant metastasis-free survival (DMFS), relapse-free survival (RFS) and overall survival (OS) (**Fig 6D–6G**) and therapeutic responses (**S7 Fig**) in breast cancer patients (see details in **S6 Text**), in consistent with previous clinical studies [46]. Moreover, both *in vitro* and *in vivo* experiments [47,48] have validated that FOXM1 plays important roles in breast cancer progression through promoting cell proliferation and cell cycle. Furthermore, FOXM1 has been used as a key drug target in breast cancer [49,50], and several drugs (e.g., daunorubicin, doxorubicin, epirubicin, and tamoxifen [51]) developed to target or inhibit FOXM1 have been tested in clinical trials (https://clinicaltrials.gov/). These evidences suggest that our network inference and analysis approach is effective to identify key genes of cancer progression or candidate drug targets.

## Validation of the FOXM1 subnetwork

A subnetwork was reconstructed for FOXM1, which predicted that FOXM1 could positively regulate ASPM, CDCA8, KIF2C, MCM10, MELK, NCAPG, SHCBP1 and STIL (**Fig 7A**). Preliminary investigation indicated that, except for STIL, the other 7 genes were functionally associated with FOXM1 according to String (https://string-db.org/), a database of functional protein-protein interaction networks (**S8 Fig**). We proceeded to validate the expression changes of these predicted target genes using microarray data of MCF-7 cells that were treated with DMSO (control) or thiostrepton (a FOXM1 inhibitor) for 6 hours (GSE40766 [42]). We found that, except for SCCBP1 and STIL, the other 6 genes were significantly downregulated after FOXM1 inhibition (**Fig 7B**). The statistical significance was assessed using Wilcoxon rank-sum test (one-tailed) *p* values. These results suggest that PROB well predicted both the directions and signs of the edges in the FOXM1 subnetwork.

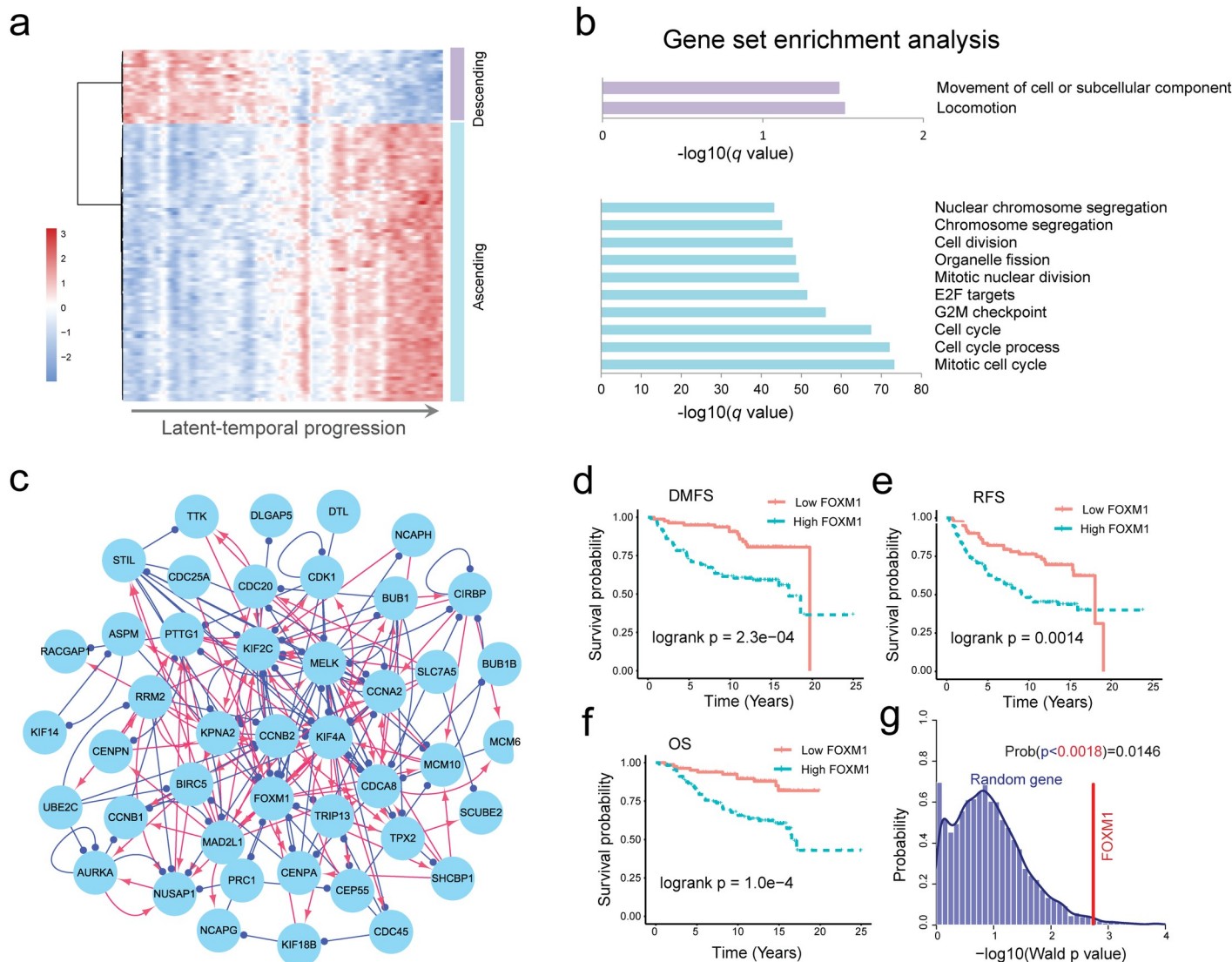

**Fig 6. FOXM1 was revealed as a key gene underlying breast cancer progression by PROB.** The gene expression data of 196 patients with clinical information (e.g., grade) were extracted from the GEO database (GSE7390 [40]). (**a**) Heatmap showing the expression profile of 100 selected genes that were most sustainably ascending (blue group) or descending (purple group) during cancer progression. (**b**) Gene set enrichment analysis for the descending genes (upper panel) and ascending genes (lower panel). The descending genes were enriched in local movement processes, and the ascending genes were mainly enriched in cell cycle and cell division processes. (**c**) The inferred GRN for the 100 genes. FOXM1 was found to be a hub gene in the network. (**d-f**) Clinical relevance of FOXM1 for breast cancer patients with respect to distant metastasis-free survival (DMFS) (**d**), relapse-free survival (RFS) (**e**) and overall survival (OS) (**f**). (**g**) Significance test of the prognostic power of FOXM1 using a bootstrapping approach. The *p* value from the permutation test was 0.0146, verifying the statistical significance of the prognostic power of FOXM1.

Moreover, we used ChIP-seq data (GSE40762 [42]) to analyze the binding of FOXM1 to the predicted targeted genes (**S7 Text**). Both estrogen-dependent ER (+) MCF-7 and estrogen-independent ER (-) MDA-MB-231 human breast cancer cell lines were used for analysis. The analysis showed that FOXM1 binds ASPM, CDCA8 and KIF2C in both cell lines (**Fig 7C–7H**). We note that the above three targets of FOXM1 were not previously reported by the widely used databases of transcriptional factor targets (e.g., TRANSFAC [52] and TRRUST v2 [53]). Interestingly, in another human mammary epithelial cell line (HMEC) (GSE62425 [54]) (**S9 Fig**), the binding of FOXM1 to CDCA8 was absent, suggesting the emerging binding of FOXM1 to certain genes during the formation of breast cancer. In addition, we confirmed that

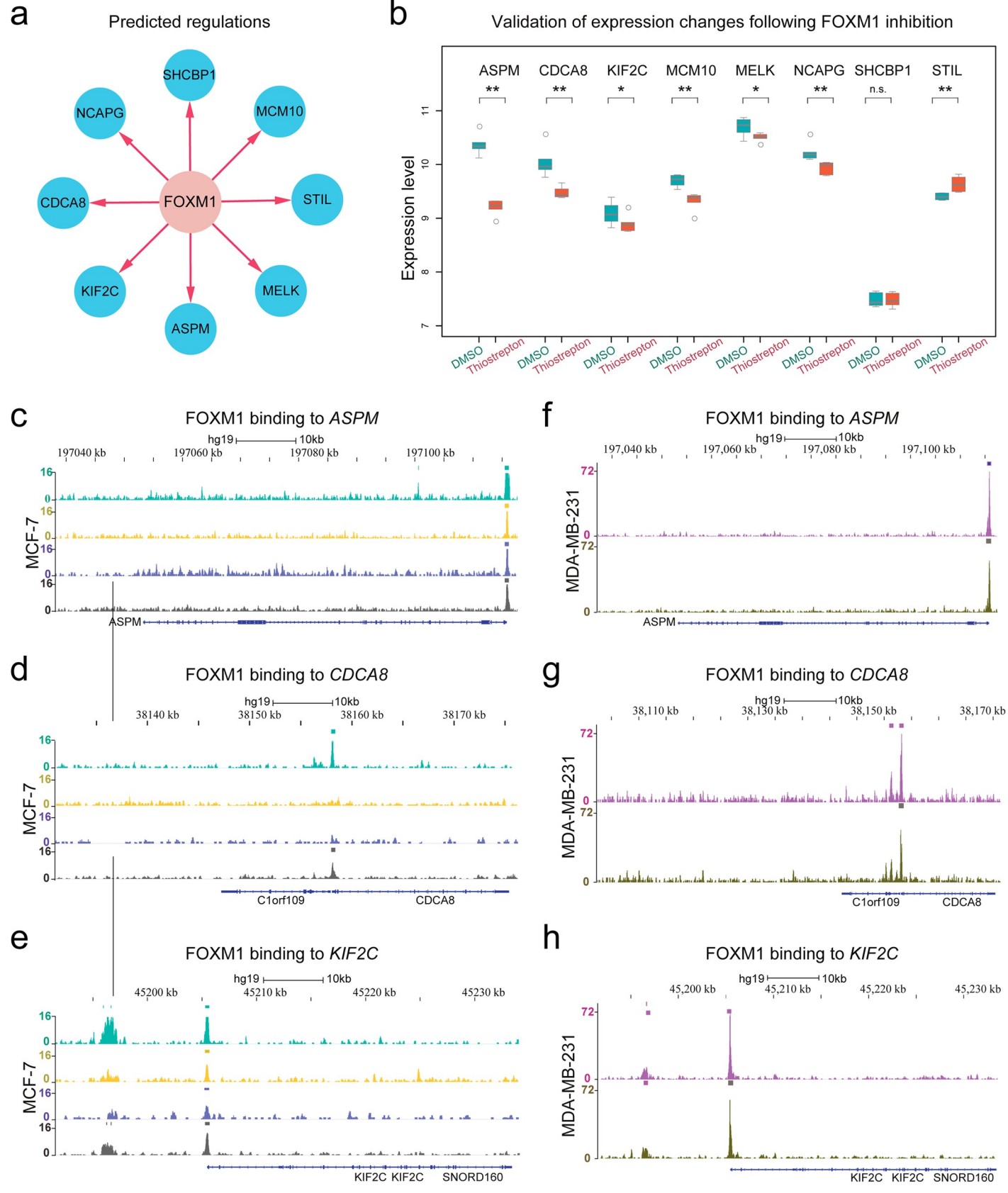

**Fig 7. Validation of the predicted FOXM1 subnetwork.** (**a**) The subnetwork of FOXM1 with predicted target genes. (**b**) Validation of the expression changes of the predicted target genes of FOXM1 with perturbation experiments. MCF-7 cells were treated with DMSO (control) or thiostrepton (a FOXM1 inhibitor) for 48 hours. Except for SCCBP1 and STIL, the other 6 genes were significantly down-regulated after FOXM1 inhibition. (**c-e**) ChIP-seq analysis of FOXM1 in the MCF-7 cell line with four biological replicates, showing that FOXM1 binds ASPM, CDCA8 and KIF2C. (**f-h**) ChIP-seq analysis of FOXM1 in the MDA-MB-231 cell line with two biological replicates, showing that FOXM1 binds ASPM, CDCA8 and KIF2C.

the expression levels of the above three genes, ASPM, CDCA8 and KIF2C, were significantly reduced following the knockdown or silencing of FOXM1 based on both microarray data in BT-20 breast cancer cells (GSE2222 [55]) (S10A–S10C Fig) and RNA-seq data in MCF-7 breast cancer cells (GSE58626 [56]) (S10D–S10F Fig). These findings suggest that FOXM1 not only positively regulates the expression of but also directly binds to some of the predicted genes.

## Discussion

PROB provides a novel tool for inferring cancer progression and GRNs from cross-sectional data. Our approach is based on a dynamical systems representation of gene interactions during cancer progression. The inverse problem with respect to GRN reconstruction was solved by combining latent progression estimation and Bayesian inference for high-dimensional dynamic systems. PROB can be used to generate experimentally testable hypotheses on the molecular regulatory mechanisms of gene regulation during cancer progression and to identify network-based gene biomarkers for predicting cancer prognosis and treatment response.

Besides cross-sectional bulk transcriptomic data, our method can be naturally applied to time-course scRNA-seq data (**Fig 3**). Although scRNA-seq data can be used to infer GRNs during cell differentiation or development, it is currently not feasible to use scRNA-seq to investigate long term cancer progression due to patient heterogeneity, difficulty in acquisition of massive samples and expensive cost. In view of this, clinical transcriptomic data of cancer patients provide an alternative way to infer GRNs underlying cancer progression. The novelty and superiority of PROB can be first attributed to the successful ordering of tumor samples by using both gene expression data and staging information. Our proposed stage-weighted Gaussian kernel allows construction of diffusion-like random walks to quantify the temporal progression distance (TPD) between two patients (Eq (8)). The diffusion map, as a manifold-based nonlinear dimension reduction method, has been recently applied to scRNA-seq data analysis [26,57–59]. One major difficulty in applying diffusion maps for inferring pseudo trajectories lies in identifying the rooting point when using scRNA-seq data itself, and it often needs additional biological knowledge. An advantage of clinical transcriptomic data is that staging or grading information is usually available for samples as well, allowing development of an algorithm that automatically identifies the rooting point (Eq (9)). We demonstrated that incorporating staging information into the temporal progression inference significantly improved its accuracy (**S1 Fig**) and that our method significantly outperformed existing pseudotime inference methods (**Figs 3B** and S6).

Considering technical variabilities in the sample-based transcriptomic data, it is important to have good robustness of the interaction coefficients in the GRN model with respect to the perturbation of the temporal progression. In addition to proving such property mathematically, through simulations we found that PROB inference of both the progression trajectory and the gene network structure are rather robust to noise in the data (**Figs 2**, S4 and S5). In addition, PROB is computationally efficient for GRN inference, which could be completed within 1 minute on the three real datasets analyzed in this study (**S4 Table**).

For clinical applications, our method can be used to identify key genes for early detection of cancer progression and design of therapeutic targets. By recovering the temporal dynamics of gene expression in terms of the disease progression, PROB provides insights into exploiting

kinetic features of functionally important genes that may be used as predictive biomarkers or drug targets. In the case study of bladder cancer progression, we have demonstrated that ACSS1 and PTNT12 played important roles in EMT during bladder cancer progression from UC to SARC and their expressions dynamically changed over the progression (**Figs 4** and **5**). Therefore, we hypothesized that the temporal dynamics of EMT regulatory genes (e.g., ACSS1 or PTPN12) could be exploited to predict cancer progression. To this end, a logistic regression model was developed to predict EMT states or histological subtypes (UC vs. SARC) of bladder cancer based on the expression levels of ACSS1 and PTPN12, which showed good predictive accuracy (**S11 Fig**). As such, the early changes in expressions of ACSS1 and PTPN12 during the progression of UC to SARC may be relevant for the early detection of SARC.

In another case study of breast cancer, FOXM1, a drugable target, was identified as a key regulator underlying breast cancer progression (**Fig 6**) and, importantly, the predicted FOXM1-target regulations were validated (**Fig 7**). Furthermore, here, we propose a GRN kinetic signature (**S8 Text**) based on FOXM1-targeted gene interactions to prognosticate relapse in breast cancer. Kaplan-Meier (K-M) survival curves were plotted for the high-risk group (green) and low-risk group (red) of patients with respect to relapse-free survival (RFS) (S12A–S12C Fig). The log-rank test $p$ values for all three datasets were less than 1e-4. Moreover, we tested the statistical significance of the FOXM1-targets interactions in predicting relapse in breast cancer using a bootstrapping approach (**S8 Text**). We compared the prognostic power (Wald test $p$ value) of the FOXM1-predicted targets with that of 10000 sets of 8 randomly selected genes. The permutation test $p$ values for all three datasets were less than 0.05 (S12D–S12F Fig), verifying the non-randomness of the predicted targeted genes of FOXM1. These results demonstrated that the predicted FOXM1-target interactions could be used as a biomarker for prognosticating relapse in breast cancer. The latent-temporal progression–based casual network reconstruction method proposed in this study will likely innovate other network-based methodologies, such as those in system genetics [60,61], network pharmacology [62,63], and network medicine [64,65].

Our method has several limitations that could be improved in future studies. For example, in the current method, only gene expression profiles and staging information from patient samples have been used for latent-temporal progression modeling. Other covariates, for example, age, genetic mutation, and molecular subtypes, might also be useful for progression inference [66]. Statistical models that integrate multiple aspects of clinical information will provide better inference of disease progression.

In summary, we have developed a novel latent-temporal progression-based Bayesian Lasso method, PROB, to infer directed and signed gene networks from prevalent cross-sectional transcriptomic data. PROB provides a dynamic and systems perspective for characterizing and understanding cancer progression based on patients' data. Our study also sheds light on facilitating the regulatory network-based approach to identifying key genes or therapeutic targets for the prognosis or treatment of cancers.

## Supporting information

**S1 Fig. Incorporation of staging information significantly improved the accuracy of latent-temporal progression inference.** We compared PROB with its several variants: '$\omega_{xy} = 1$' represents setting the weight coefficient $\omega_{xy}$ in Eq (2) to be 1; '$x_{ref}$ = random' represents randomly assigning the reference point (Eq (9)) to identify the rooting point as the previous pseudotime inference methods usually did; 'No stage' represents leaving out the stage information, i.e., both '$\omega_{xy} = 1$' and '$x_{ref}$ = random'. Kendall tau correlation coefficient or determinant coefficient ($R^2$) between the inferred temporal progression and the staging data (or the capture time

in scRNA-seq data) was calculated for each method. (**a**) Kendall tau for the TCGA COAD dataset. (**b**) $R^2$ for the TCGA COAD dataset. (**c**) Kendall tau for the TCGA SKCM dataset. (**d**) $R^2$ for the TCGA SKCM dataset. (**e**) Kendall tau for the GSE7390 dataset. (**f**) $R^2$ for the GSE7390 dataset. (**g**) Kendall tau for the LPS scRNA-seq dataset. (**h**) $R^2$ for the LPS scRNA-seq dataset.
(TIF)

**S2 Fig. Illustration of PROB using a synthetic dataset.** (**a**) A set of synthetic gene expression data of 100 cancer patients along with true progression. For illustration and visualization purpose, only 6 genes were tested. (**b**) Simulated tumor sample-based gene expression data by randomizing sample IDs of data in (a) but retaining staging information, which was used as input for PROB. (**c**) Comparison of inferred temporal progression with true progression in the synthetic dataset. (**d**) Recovered gene expression dynamics along with temporal progression. (**e**) The inferred GRN using Bayesian LASSO method based on data in (d). (**f**) Accuracy of GRN inference evaluated using area under curve (AUC) of ROC for the inferred network compared to the ground-truth network (AUC = 0.8395).
(TIF)

**S3 Fig. Posterior distribution of regulatory parameters associated with GRN inference in S1 Fig.** The sub-figure located in *i*-th row and *j*-th column represents the posterior distribution of the regulatory coefficient from gene *j* (Gj) to gene *i* (Gi). The red lines represent the parameter values of $a_{ij}$ used for generating the ground-truth network as in Equation (S22). An interaction was viewed present if the *k*% credible interval for corresponding regulatory coefficient $a_{ij}$ did not contain zero, otherwise absence.
(TIF)

**S4 Fig. Evaluation indexes for temporal progression inference and GRN inference under different variability levels in the synthetic datasets.** The levels of measurement variabilities in the synthetic data were quantified using the coefficient of variations (CVs) (from 0% to 30%). (**a-b**) Root of mean squared error (RMSE) and Spearman correlation used for evaluating the accuracy of the temporal progression inference. (**c-f**) AUC, accuracy rate, positive predictive rate (PPV) and Matthews correlation coefficient (MCC) used for evaluating the robustness of the GRN inference.
(TIF)

**S5 Fig. Testing the robustness of PROB against exponential noises.** The noises were generated from the exponential distribution with mean ranging from 0 to 0.3. (**a**) Kendall correlation for evaluating the accuracy of the temporal progression inference. (**b**) AUC for evaluating the robustness of the GRN inference.
(TIF)

**S6 Fig. Benchmarking PROB with other existing pseudotime inference methods.** A set of scRNA-seq data of dendritic cells stimulated with LPS was used for benchmarking. The cells were sequenced at 1, 2, 4 and 6h after stimulation of LPS. We compared PROB with other pseudotime inference methods (Slice, Slicer, PhenoPath, Wishbone, PAGA, Monocole2, DPT, Tscan) in cell ordering. The coefficient of determination (i.e., $R^2$) between the estimated pseudotime and the capture time of cells was used for evaluation. PROB outperformed the other existing methods.
(TIF)

**S7 Fig. FOXM1 expression was associated with the therapeutic responses of breast cancer patients.** Breast cancer patients who received endocrine therapy (**a**) or chemotherapy (**b**) were

included into the K-M survival analysis. Kaplan-Meier Plotter (http://kmplot.com) [67] was employed to perform analysis. Log-rank test p-value was used to assess the prognostic significance.
(TIF)

**S8 Fig. The functional interaction network of FOXM1 extracted from String database.** The network shows the co-expression or regulation between FOXM1 and the predicted targeted genes. Among 8 targeted genes of FOXM1 predicted from PROB, 7 genes (including KIF2C, SHCBP1, CDCA8, NCAPG, ASPM, MELK and MCM10) were supported by the database information.
(TIF)

**S9 Fig. ChIP-seq analysis of FOXM1 in human mammary epithelial cells (HMEC).** ChIP-seq data were downloaded from GEO database (GSE62425) [54]. The analysis results showed FOXM1 binds ASPM and KIF2C.
(TIF)

**S10 Fig. Another validation for the predicted regulation of KIF2C, ASPM and CDCA8 by FOXM1.** Microarray data and RNA-seq data on two breast cancer cell lines (BT-20 and MCF-7, respectively) were used for analyses. (**a-c**) The expression levels of the above three genes in BT-20 breast cancer cells under FOXM1 siRNA or control (mock transfection and GFP siRNA) conditions were analyzed using a set of microarray data (GSE2222) [55] (**d-f**) RNA-seq data (GSE58626) [56] of MCF-7 breast cancer cells was used to analyze the differential expressions of the above three genes after FOXM1 inhibition by using small molecule compound IB that specifically inhibits FOXM1 [56]. The knockdown or silence of FOXM1 significantly reduced the expressions of the above three genes. Wilcoxon rank sum test (one-tailed) p value was calculated to assess the statistical significance.
(TIF)

**S11 Fig. ACSS1 and PTPN12 are predictive of EMT and progression of UC to SARC.** A logistic regression model was developed to predict (**a**) EMT states or (**b**) histological subtypes (UC vs. SARC) of bladder cancer based on the expression levels of ACSS1 and PTPN12. The samples were randomly divided into training set (n = 56) and test set (n = 56). The AUCs for the EMT phenotype prediction and subtype prediction are 0.8054 and 0.9405, respectively.
(TIF)

**S12 Fig. A GRN kinetic signature predicts relapse in breast cancer.** The kinetic features of the FOXM1-target interactions were formulated as a risk score to predict relapse for breast cancer patients in multiple independent cohorts. (**a-c**) Prognostic significance of the FOXM1--target interactions with respect to predicting relapse-free survival (RFS) in breast cancer evaluated on different datasets (GSE2990 [68], GSE12093 [69] and GSE5327 [70]). The log-rank test *p* value was used to assess the statistical significance of the difference between the Kaplan-Meier (K-M) survival curves of the high-risk group (green) and the low-risk group (red) of patients. (**d-f**) Nonrandomness test of the FOXM1-target interactions in predicting relapse in breast cancer using a bootstrapping approach (**Text S6**). The permutation test *p* values for all three datasets (0.0073, 0.0401 and 0.006, respectively) were less than 0.05, verifying the statistical significance of the prognostic power of the FOXM1-target interactions.
(TIF)

**S1 Table. Out-degree values of the genes in the UC-specific and SARC-specific networks.**
(DOCX)

**S2 Table. The siRNA sequence used in this study.**
(DOCX)

**S3 Table. The specific primers used in this study.**
(DOCX)

**S4 Table. Runtime of PROB on different datasets.**
(DOCX)

**S1 Text. Progression-dependent dynamic modeling of the GRN.**
(DOCX)

**S2 Text. Proof of the Theorem 1.**
(DOCX)

**S3 Text. Implementation of PROB.**
(DOCX)

**S4 Text. Simulation study.**
(DOCX)

**S5 Text. PROB applied to real datasets.**
(DOCX)

**S6 Text. Clinical relevance of FOXM1 to breast cancer.**
(DOCX)

**S7 Text. Validation of the predicted FOXM1-targets interactions.**
(DOCX)

**S8 Text. GRN kinetic signature.**
(DOCX)

**S1 Data. Numerical data underlying graphics.**
(XLSX)

## Acknowledgments

We would like to acknowledge Profs. Tianshou Zhou, Jinzhi Lei, Yong Wang for valuable discussion. We would also like to acknowledge Drs. Zifeng Wang and Dongliang Leng for processing the Chip-seq data.

## Author Contributions

**Conceptualization:** Xiaoqiang Sun.

**Data curation:** Xiaoqiang Sun.

**Formal analysis:** Xiaoqiang Sun, Qing Nie.

**Funding acquisition:** Xiaoqiang Sun, Qing Nie.

**Investigation:** Xiaoqiang Sun, Qing Nie.

**Methodology:** Xiaoqiang Sun.

**Project administration:** Xiaoqiang Sun.

**Resources:** Xiaoqiang Sun.

**Software:** Xiaoqiang Sun.

**Supervision:** Xiaoqiang Sun.

**Validation:** Xiaoqiang Sun, Ji Zhang.

**Visualization:** Xiaoqiang Sun, Qing Nie.

**Writing – original draft:** Xiaoqiang Sun.

**Writing – review & editing:** Xiaoqiang Sun, Qing Nie.

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
