## [Decision Letter · Decision Letter 0]

12 Jan 2021

Dear Dr. Sun,

Thank you very much for submitting your manuscript "Inferring latent temporal progression and regulatory networks from cross-sectional transcriptomic data of cancer samples" for consideration at PLOS Computational Biology. As with all papers reviewed by the journal, your manuscript was reviewed by members of the editorial board and by several independent reviewers. The reviewers appreciated the attention to an important topic. Based on the reviews, we are likely to accept this manuscript for publication, providing that you modify the manuscript according to the review recommendations.

Please make sure that your code and data needed to reproduce the results of the paper are accessible at a valid URL.

Sincerely,

Sushmita Roy, Ph.D.

Deputy Editor

PLOS Computational Biology

Jian Ma

Deputy Editor

PLOS Computational Biology

[LINK]

Reviewer's Responses to Questions

**Comments to the Authors:**

Reviewer #1: Gene regulatory network (GRN) reconstruction methods usually require time-course gene expression data or perturbation experiment data. However, most available omics data from cross-section studies of cancer patients often lack temporal information, making it challenging to reconstruct the GRN for the studied cancer progression. In this work, Sun et al. present a latent-temporal progression-based bayesian method, PROB, for the inference of GRNs from the cross-sectional transcriptomic data of tumor samples. Compared with other GRN methods, the proposed method has a number of advantages: (1) First, it was designed specifically to reconstruct GRNs from tumor samples, which utilizes a lot of cancer-specific information (staging) that are not utilized by others general methods (such as GENIE3). (2) Second, the authors mathematically proved the robustness of the proposed methods. The robustness is especially important in reconstructing GRNs from cancer samples, which are usually highly noisy. Third, the proposed method requires minimal prior knowledge. For example, in other GRN reference methods, the root (node) is usually needed to pick out by the users based on their prior knowledge. The proposed methods can automatically learn the root using a proposed distance-based approach. This significantly improves the usability of the proposed method. Overall, I think that this is a fantastic method and well fit the authorship of the journal. I highly recommend the publication of this manuscript. I only have a few minor comments :

(1) In equation (5), the authors described that the root is determined as the patient (x) with the largest distance to the patients the maximal grade score (e.g., stage 4). I would suggest limiting the candidate x \\ patients with the smallest grade (e.g., grade 0). If not, theoretically, it is likely that x0 will come from a later stage (e.g., stage 2 or even stage 4 (e.g., because of a few outliers that produce a considerable distance). In that case, the root x0 will come from stage 2 or stage 4, which would not make much sense to me. As the stage information is available, I would suggest to narrow down the selection of x0 among patients in stage 0

(2) In this work, the authors simulate Gaussian noise to examine the robustness of the method. I am amazed by the robustness of the method to Gaussian noise. However, the Gaussian noise is usually well dealt with. Did the authors also simulate other types of noise (e.g., drop-out noise is also a very common noise type)? It would be great if the methods can also be tested against other noises.

Reviewer #2: General Comments

Authors developed a latent-temporal progression-based Bayesian method, PROB, for inferring gene regulatory networks (GRNs) from the cross-sectional transcriptomic data of tumor samples. Mathematical proofs and numerical verification are provided to support the robustness of PROB to the measurement variabilities in the data. Benchmarking PROB with alternative methods of GRN inference shows advantages of PROB in both pseudotime inference and GRN inference. Authors also demonstrated the applications of PROB for identification of key regulators of cancer progression or drug targets as well as performed validations experiments. Overall, the manuscript is clear and accessible.

Specific comments

Pages 9-12: Section “Latent-temporal progression-based Bayesian (PROB) method to infer GRN”

This section can be rewritten more compactly to improve reading experience. Either put all details in the main text or keep only the main idea in the main text and leave the details in the supplementary text. Current main text is an abridged version of Text S1 and does not read consistent.

Page 9: “… the root was automatically identified with the aid of staging information.”

What is “root” meant here and in the subsequent context? Does it adapt a generally used and accepted mathematical meaning? Should it be called “optimizer”?

Page 11: “The above model is then transformed into a linear regression model …”

I think it’s better phrased as “a linear system”. Rigorously speaking Y is not linear in X.

Page 12: “Therefore, the above theorem theoretically guarantees the consistency and robustness of the estimates of the regulatory coefficients.”

It can be elaborated more the role of Theorem 1; for example, add 2-3 lines of equations.

Page 12: pseudo-code of PROB

Variable E is not defined in the main text.

Definition of phi should be readdressed in the algorithm.

Variable/function PPD is not defined.

Page 13: “To this end, we defined an outgoing causality score (OCS) …”

Does OCS have any mathematical/statistical meaning?

Page 21: “The code for PROB is available at https://github.com/SunXQlab/PROB.”

This page does not exist.

Page 28: Figure 2d

Why does not AUC decrease monotonically with CV?

Page 32: Figure 6: Details related to survival analysis are missing (e.g. how authors define

high and low risk groups. What is their cut-off?) Also role of FOXM1 on breast cancer progression is known and citations are missing.

Page S10: “The codes are available at https://github.com/dongbusun/PROB.”

This page does not exist.

Page S15: “PROB applied to realistic datasets”

Realistic -> real

**Have all data underlying the figures and results presented in the manuscript been provided?**

Reviewer #1: Yes

Reviewer #2: **No: **

PLOS authors have the option to publish the peer review history of their article (what does this mean?). If published, this will include your full peer review and any attached files.

Reviewer #1: No

Reviewer #2: No
---

## [Decision Letter · Decision Letter 1]

15 Feb 2021

Dear Dr. Sun,

We are pleased to inform you that your manuscript 'Inferring latent temporal progression and regulatory networks from cross-sectional transcriptomic data of cancer samples' has been provisionally accepted for publication in PLOS Computational Biology.

Please provide the numerical data underlying graphs described in the paper. 

Best regards,

Sushmita Roy, Ph.D.

Deputy Editor

PLOS Computational Biology

Jian Ma

Deputy Editor

PLOS Computational Biology

Reviewer's Responses to Questions

**Comments to the Authors:**

Reviewer #2: Authors addressed my concerns

**Have all data underlying the figures and results presented in the manuscript been provided?**

Reviewer #2: **No: **Not all numerical data that underlies graphs or summary statistics provided

PLOS authors have the option to publish the peer review history of their article (what does this mean?). If published, this will include your full peer review and any attached files.

Reviewer #2: No

---

## [Editor Report · Acceptance letter]

28 Feb 2021

PCOMPBIOL-D-20-01798R1 

Inferring latent temporal progression and regulatory networks from cross-sectional transcriptomic data of cancer samples

Dear Dr Sun,

I am pleased to inform you that your manuscript has been formally accepted for publication in PLOS Computational Biology. Your manuscript is now with our production department and you will be notified of the publication date in due course.

With kind regards,

Alice Ellingham
